# Validity and Reliability of the Orthelligent Pro Sensor for Measuring Single-Leg Vertical Jump Height in Healthy Athletic Adults

**DOI:** 10.3390/s24123699

**Published:** 2024-06-07

**Authors:** Caterina Pasquale, Pierrette Baschung Pfister, Manuel Kuhn, Thomas Stöggl

**Affiliations:** 1Department of Physiotherapy and Occupational Therapy, University Hospital Zurich, 8091 Zurich, Switzerland; 2Directorate of Research and Education, Physiotherapy Occupational Therapy Research Center, University Hospital Zurich, 8091 Zurich, Switzerland; pierrette.baschung@usz.ch; 3Faculty of Medicine, University of Zurich, 8006 Zurich, Switzerland; manuel.kuhn@usz.ch; 4Department of Pulmonology, University Hospital Zurich, 8091 Zurich, Switzerland; 5Department of Sport and Exercise Science, University of Salzburg, 5020 Salzburg, Austria; thomas.stoeggl@plus.ac.at; 6Red Bull Athlete Performance Center, 5303 Salzburg, Austria

**Keywords:** wearable sensor, single-leg countermovement jump, assessment, jumping ability, return to sport, test–retest

## Abstract

The Orthelligent Pro sensor is a practicable, portable measuring instrument. This study assessed the validity and reliability of this sensor in measuring single-leg countermovement jumps. Fifty healthy athletic adults participated in two measurement sessions a week apart in time. They performed single-leg countermovement jumps on the force plate while wearing the Orthelligent Pro sensor on their lower leg. During the first measurement session, Tester 1 invited the participants to make three single-leg countermovement jumps; subsequently, Tester 2 did the same. For assessing the sensor’s intratester reliability, Tester 1 again invited the participants to make three single-leg countermovement jumps during the second measurement session. The sensor’s validity was assessed by using the force plate results as the gold standard. To determinate the agreement between two measurements, Bland–Altman plots were created. The intertester reliability (ICC = 0.99; 0.97) and intratester reliability (ICC = 0.96; 0.82) were both excellent. The validity calculated (i) on the basis of the mean value of three jumps and (ii) on the basis of the maximum value of three jumps was very high, but it showed a systematic error. Taking this error into account, physiotherapists can use the Orthelligent Pro sensor as a valid and reliable instrument for measuring the jump height of countermovement jumps.

## 1. Introduction

In various jump-intensive sports like volleyball and basketball, the maximum height of a vertical jump significantly impacts performance [1]. The vertical jump test is commonly employed to assess lower extremity explosiveness and inform training methodologies [2,3,4]. Given the prevalence of injuries during single-leg motor tasks, the limb symmetry index becomes crucial in rehabilitation, guiding decisions on returning to sport safely [5]. A limb symmetry index value exceeding 90% compared to the healthy leg is indicative of readiness for return to sport [6], with the one-legged vertical jump showing strong correlations with isokinetic strength tests and the return to sport scale after anterior cruciate ligament reconstruction [7].

While measuring jump height is essential, traditional methods like 3D motion analysis and force plates, though considered criterion standards, are often prohibitively expensive and require substantial space [8,9,10]. As a result, healthcare providers and coaches may face challenges in practical implementation. Fortunately, cost-effective alternatives such as the MyJump App [11], Jumpo app [12], and Vertec [13] offer accessible solutions for measuring jump height. Brooks et al. [14] compared the “MyJump App” with the force plate and reported a very high validity (r = 0.98). Also the Jumpo app shows very high validity (r = 0.93–0.96) for the vertical jump height compared to the force plate [12]. 

Recently, the Orthelligent Pro sensor has emerged as a promising tool for assessing various functional movements of the lower extremities, providing a cost-effective and practical means of evaluating return to sport readiness [15]. To the best of our knowledge, up to now no study has tested the validity and reliability of the Orthelligent Pro sensor for measuring vertical jump height. Our study aims to evaluate the reliability and validity of Orthelligent Pro sensor for measuring vertical jump height with a single-leg countermovement jump (CMJ) in healthy athletic adults, particularly in comparison to the criterion method of force plate analysis.

We hypothesize that the Orthelligent Pro sensor will demonstrate excellent intra- and intertester reliability (ICC > 0.75), with a smallest detectable change (SDC) below 20% and a standard error of measurement (SEM) below 10%. Additionally, we expect it to exhibit a good concurrent validity (high or very high, r > 0.7) with the criterion standards, showing no systematic differences in jump height measurements.

## 2. Materials and Methods

A test–retest cross-sectional study with two testers was performed. The Ethics Committee of the Canton of Zurich approved the study (BASEC No. 2021-D0004).

### 2.1. Participants and Recruitment

Participants were recruited among medical staff at the University Hospital Zürich. Potential participants received information about the aim of the study and the risks associated with participation. After screening for inclusion and exclusion criteria and in the case of mutual consent, the participants signed an informed consent form. Following the recommendation of Cosmin [16] for testing reliability, 50 persons were selected. The inclusion criteria were male/female, aged between 18 and 65 years, active in sports (at least once per week, jumping elements during training), familiar with CMJs, knowledge of the German language, and the ability to understand verbal and written instructions. The exclusion criteria were known pregnancy, drug or alcohol abuse, and an inability to follow the trial procedures (e.g., due to language problems, mental illness, or dementia). Additionaly, persons with lower limb injuries during the last year, concomitant diseases of the foot, knee, or hip, tiring lower limb training the day before, or strenuous activity of the lower extremity one hour before the test were excluded. 

### 2.2. Measuring Instruments

#### 2.2.1. Orthelligent Pro

By means of the sensor holder, the Orthelligent Pro sensor was attached to the thickest part of the calf on the participant’s dominant leg. The sensor (2.0, firmware v2.982, Oped, Valley/Oberlaindern, Germany) calculates jump height via flight time. The integrated acceleration sensor (200 Hz) and a rotation speed sensor detected the time when the heel lost contact with the ground and the measurement began. The measurement data were visible on the Orthelligent Pro app (software version: 2.14.0).

#### 2.2.2. Force Plate

The participants performed the vertical jump on a Kistler piezoelectric 1D force plate (920 × 920 × 125 mm; type 9290DD; capture rate: 500 Hz; Winterthur, Switzerland) [17]. The Kistler force plate was connected to a PC running the Kistler Mars Quattro Jump software (v5.3.0.245). The start of the jump motion (start of downward motion during CMJ) and the jump-off (takeoff from the force plate) triggers for calculation of the force impulse were automatically set based on the Kistler MARS Software producing a standard set of biomechanical jump metrics. A previous study by Hébert-Losier and Beaven [18] demonstrated an acceptable between-day reproducibility for Squat Jump and CMJ (ICC 0.88 and 0.84).

### 2.3. Procedures

Before testing, the participants received verbal instructions about the measurement procedure. Every participant warmed up on a bicycle ergometer for five minutes. Warm-up was standardized at 50–70 W (Borg RPE Scale = 11 [RPE = Rating of Perceived Exertion]) on the bicycle ergometer (Kardiomed basic cycle, Proxomed, Steckborn, Switzerland). The tester verbally instructed the participants and demonstrated the single-leg CMJ once. Every participant performed three test jumps.

During the first visit, the participants were tested twice. At the first visit, Tester 1 instructed and measured three single jumps. The sensor was removed and Tester 2 reattached it on the same leg. Afterwards, Tester 2 instructed and measured three single jumps. 

At the second visit, performed within one week, Tester 1 instructed and measured three single jumps. The first visit lasted 30 min per participant, the second 15 min. The test procedure is illustrated in Figure 1.

Tester 1 was the first author (female, 31 years old, BSc physiotherapist with eight years of professional experience). Tester 2 was female, 26 years old, pre-trainee BSc occupational therapist without therapeutic work experience. Tester 1 trained Tester 2 in advance for the measurements.

### 2.4. Single-Leg CMJ

The participants stood single-legged, on their takeoff leg, and barefoot on the force plate with their hands on their hips. They started in an extended position, performed a single-leg CMJ, jumped as high as possible, and landed on the same leg (see Figure 1D) [15]. Stopping in the deep squat position was not allowed, otherwise the jump had to be repeated. The jumping leg had to remain extended during the flight phase.

### 2.5. Data Collection

For each participant, the testers entered all trial-relevant data in a case report form (CRF): consent, inclusion and exclusion criteria, demographic data, measured values, and information on adverse events. The Orthelligent Pro data were stored in the associated app, and force plate data were saved in the Kistler Mars Quattro Jump software [17].

### 2.6. Data Analysis and Evaluation

For statistical analysis, IBM SPSS version 25.0 software (SPSS, Inc., Chicago, IL, USA) was used. By means of R version 4.0.3 (R Core Team 2020, R Foundation for Statistical Computing, Vienna, Austria), Bland–Altman plots were created [19]. For reliability analysis, ICC (two-way-random; absolute agreement) was calculated. Reliability was assessed based on (i) the mean value of three jumps (ICC of the average measurement) and (ii) the maximum value of three jumps, independent of trial number (ICC of individual measurements). The categorizations for reliability were 0–0.4 = low, 0.4–0.75 = sufficient to medium, 0.75–1 = excellent [20]. To complement the correlation, Bland–Altman plots were created. In order to determine the amount of the agreement between repeated measurements, the standard error of measurement (SEM) ($SEM=\sigma \times \sqrt{1-ICC}$) was calculated [21]. The SEM represents the standard deviation of the repeated measurements of a test person. Additionally, the smallest detectable change (SDC) ($SDC=1.96\times \sqrt{2}\times SEM$) was calculated [21].

To assess validity, correlation was determined using Pearson’s correlation with force plate data based on (i) the mean value of three jumps and (ii) the maximum value of three jumps. The correlation coefficients were interpreted as follows: r 0–0.25 = none or very low, r 0.26–0.49 = low, r 0.50–0.69 = moderate, r 0.70–0.89 = high, r 0.90–1.0 = very high [22]. In addition, differences between the results of the Orthelligent Pro sensor and the force plate data were analyzed using Student’s *t*-test.

## 3. Results

Fifty patients aged 20–55 years participated in this study. Their anthropometric data are shown in Table 1. Four participants could not attend the second visit. Therefore, 46 participants were included for measuring intratester reliability. At Visit 2 (Measurement 3), one patient dropped out due to very low and obviously unrealistic measured values. To determine validity and intertester reliability, 100 single-leg CMJs of 50 participants were compared. To assess intratester reliability, 90 jumps of 45 participants were analyzed (Figure 2).

### 3.1. Intertester Reliability 

The intertester reliability was excellent (based on the calculation of the mean and the maximum of three jumps) (Table 2). The SEM and SDC were higher for the maximum of three jumps (0.80 versus 0.46 and 2.21 versus 1.28). The Bland–Altman plots showed no systematic error and no systematic difference (see Figure 3). The SEM and SDC were very small. The variance of the differences was independent from the measured values. 

### 3.2. Intratester Reliability 

The Orthelligent Pro sensor revealed a very good intratester reliability based on the calculation of the mean and a good reliability based on the calculation of the maximum (Table 3). The SEM and SDC were higher for the mean than for the maximum of three jumps. The Bland–Altman plots indicated a small, non-significant (mean *p* = 0.084, maximum *p* = 0.078) systematic error in both evaluations (see Figure 4).

### 3.3. Validity of the Orthelligent Pro Sensor

The Pearson correlation for validity showed a very high agreement (r > 0.94, *p* < 0.001) between the Orthelligent Pro sensor and the force plate in all evaluations (based on the calculation of the mean and the maximum; see Table 4). The Bland–Altman plots revealed a systemic error. This is reflected in the *t*-test showing significant differences (see Table 4). The values of the Orthelligent Pro sensor were higher than those of the force plate. There was a mean difference of about 4 cm to the force plate (based on the calculation of the mean and the maximum; see Figure 5). 

## 4. Discussion

The results confirmed our hypothesis that the Orthelligent Pro sensor shows an excellent inter- (ICC = 0.97–0.99) and intratester reliability (ICC 0.82–0.96). This relates to the calculation of the mean of three jumps, as well as to the calculation of the maximum of three jumps. However, the hypotheses for the SEM und SDC values were only confirmed for intertester reliability (SEM 3–5%; SDC 9–14%) and not for intratester reliability (SEM 10–16%; SDC 28–45%). We can confirm the hypothesis that the measurement of jump height by means of the Orthelligent Pro sensor correlates very highly with the force plate to calculate the mean of three jumps and the maximum of three jumps (r = 0.94–0.96). After careful analysis, we have identified a rounded systematic error of 4 cm. This error was evaluated using the mean of three jumps, as well as in calculating the maximum of three jumps.

The intertester reliability values of this study are in accordance with the values of other measurement tools. The intertester reliability of the “MyJump App” showed very high values (ICC = 0.999) with a mean difference of 0.1 cm [23]. In the current study, the intertester reliability of the mean value was ICC = 0.99 and the maximum value was ICC = 0.97 (mean difference = 0.02 cm). In our study, the intratester reliability was excellent (ICC = 0.82; 0.96). In comparison, Stanton et al. [11] reported ICC = 0.99 for the “MyJump App”. However, at the second measurement point one week later there was a mean difference of 0.4 cm. At the first measurement point, the test persons jumped higher. We also identified a mean difference of 0.3 cm and 0.7 cm between the measured values of the first and the third measurement points; however, this is clinically negligible. From this, we can conclude that jump height is difficult to measure in a test–retest setting, although the inclusion criteria such as no tiring lower limb training and no strenuous activity of the lower extremity one hour before were strictly observed. It seems that many factors affect measurements on different days.

The systematic error in our study can be explained by different measurement methods. The Orthelligent Pro sensor measures jump height with an accelerometer and calculates jump height via flight time. Systematic error has been a common issue in previous studies addressing jump height calculation. Attia et al. [24] compared “Optojump” (calculations via flight time) with the force plate (calculations via ground reaction force). They reported a high correlation (ICC = 0.994) but a systematic error of 11 cm. Brooks et al. [14] also compared the “MyJump App” (calculations via flight time) with the force plate (via ground reaction force) and reported a very high validity (r = 0.98). However, they identified a systematic error of 3 cm. This is comparable to our results. 

In practice, you should be aware that the measurement methods are different. Therefore, a comparison of measured values can only be made with the same measuring instrument and the values cannot be compared with values from another instrument. For calculations based on the mean of three jumps, the intratester reliability revealed a higher correlation than for calculations based on the maximum. For other performance tests, it is also recommended to take several repetitions and use the mean for better reliability [25,26]. The theory also describes that the average of measurements leads to higher reliability [21].

In summary, for clinicians, it is crucial to consistently use the same device and measurement technique (flight time or ground reaction force) to ensure optimal validity. When comparing the Orthelligent Pro sensor with the criterion standard, there is a notable systematic error to consider. Additionally, utilizing the mean of three jumps proves to be a more reliable method than relying on the maximum value from three jumps. Lastly, the person conducting the measurements appears to have minimal impact, as the intertester reliability remained satisfactory. Furthermore, the Orthelligent Pro sensor can reliably measure jump height with a systematic error of approximately 4 cm, making it a valuable gadget for clinicians. It is particularly useful for measuring the jump height of a single leg, and thus, it plays an important role in the clinical assessment of the limb symmetry index. However, it is essential to emphasize that this index must undergo validation in future studies to ensure its reliability and accuracy in clinical settings.

Our study has some limitations. First, participants were healthy athletic volunteers and the activity level was only defined by being active at least once a week. Therefore, transferring the results to patients needs to be further elaborated. Second, no participant jumped higher than 30 cm. Thus, the results need to be performed in persons with a higher vertical jump ability (e.g., >30 cm jump height). A future study should address these aspects. Third, the sampling frequency of the Orthelligent Pro sensor was only 200 Hz. Thus, the possible influence of a lower sampling rate can create a difference in the measurement. Fourth, measuring reliability on the basis of maximum jump height proved to be difficult. Even with the instruction to “jump as high as possible”, it was difficult to exactly equalize the bias of pre-fatigue and the participants’ physical condition after one week. This may have influenced the intratester reliability. Fifth, 1% of the Orthelligent Pro sensor measurement values appeared to be unrealistic, i.e., very high or very low. This problem was particularly evident at the third measurement point. The cause remains unclear. Nevertheless, we included these values in our analysis. In practice, instructors could ask the person to repeat the jump in the case of implausible measurement values.

## 5. Conclusions

This study demonstrates the excellent intra- and intertester reliability of the Orthelligent Pro sensor for measuring vertical jump height in healthy athletic persons. Relying on the mean value of three jumps is recommendable in order to achieve a higher reliability for assessments in daily sports clinician practice. However, we have to consider the relatively high threshold (4–7 cm) for the SDC in the test–retest context. While the validity compared to the criterion-standard force plate is notably high, systematic errors in the Orthelligent Pro sensor measurements must be acknowledged in clinical settings. Despite this, the Orthelligent Pro sensor proves to be a cost-effective, practical, and portable tool for single-leg CMJ measurements, emphasizing the importance of consistent instrument use. Future research should explore reliability in target patient populations to ensure applicability.

## Figures and Tables

**Figure 1 sensors-24-03699-f001:**
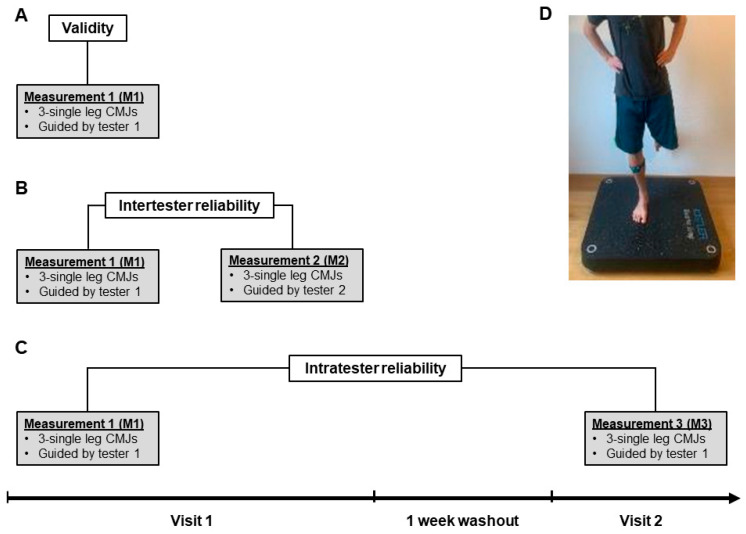
Experimental setup illustrating the validity, intertester, and intratester reliability assessments. (**A**) Validity determined using simultaneously recorded single-leg counter movement jumps at Measurement 1 comparing the Orthelligent Pro sensor with force plate (ground reaction force). (**B**) Intertester reliability comparing Measurement 1 with Measurement 2. (**C**) Intratester reliability assessed by comparing Measurement 1 with Measurement 3. (**D**) Study setup depicting the participant in the starting position for the single-leg CMJ on the force plate while wearing the Orthelligent Pro sensor around the lower leg.

**Figure 2 sensors-24-03699-f002:**
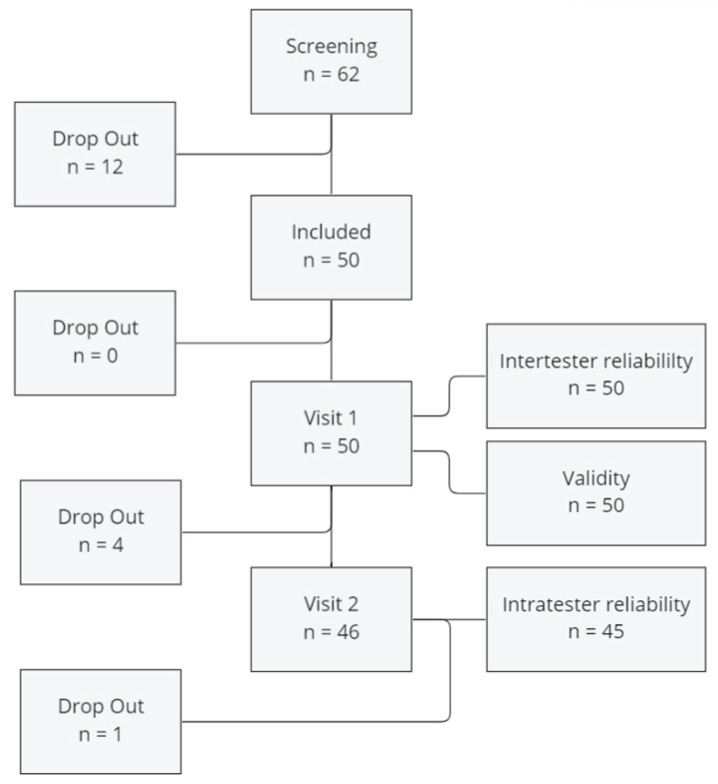
Flow chart.

**Figure 3 sensors-24-03699-f003:**
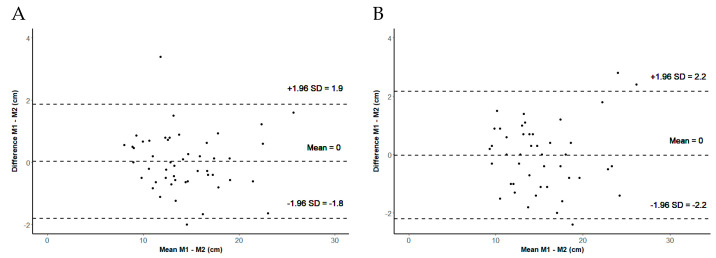
Intertester reliability. (**A**) Bland–Altman plot illustrating the mean difference (mean) and the 95% confidence interval of jump height between Measurement 1 (M1) and Measurement 2 (M2), based on the means of three jumps; (**B**) Bland–Altman plot illustrating the mean difference (mean) and the 95% confidence interval of jump height between Measurement 1 (M1) and Measurement 2 (M2), based on the maximum of three jumps.

**Figure 4 sensors-24-03699-f004:**
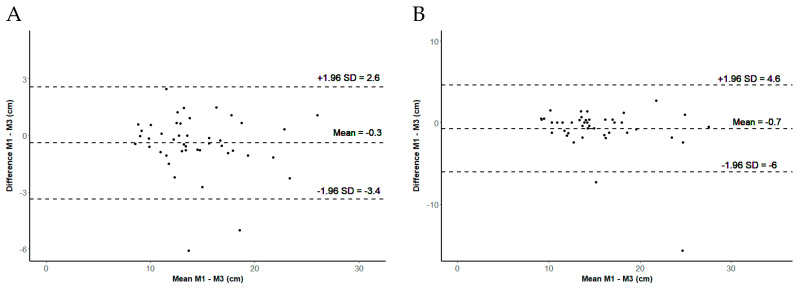
Intratester reliability. (**A**) Bland–Altman plot illustrating the mean difference (mean) and the 95% confidence interval of jump height between Measurement 1 (M1) and Measurement 3 (M3), based on the means of three jumps; (**B**) Bland–Altman plot illustrating the mean difference (mean) and the 95% confidence interval of jump height between Measurement 1 (M1) and Measurement 3 (M3), based on the maximum of three jumps.

**Figure 5 sensors-24-03699-f005:**
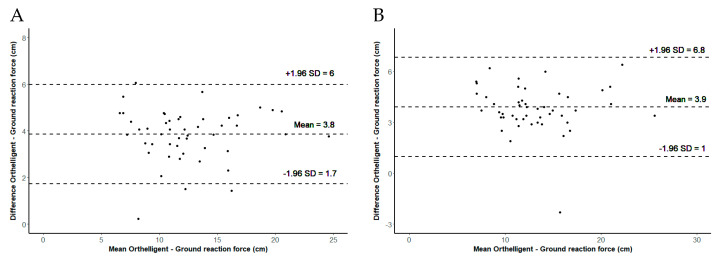
The Orthelligent Pro sensor versus force plate (calculated via ground reaction force). (**A**) Bland–Altman plot illustrating the mean difference (mean) and 95% confidence interval of jump height measured by means of the Orthelligent Pro sensor or force plate (calculated via ground reaction force), based on the mean of three jumps; (**B**) Bland–Altman plot illustrating the mean difference (mean) and 95% confidence interval of jump height measured by means of the Orthelligent Pro sensor or force plate (calculated via ground reaction force), based on the maximum of three jumps.

**Table 1 sensors-24-03699-t001:** Anthropometric data of the participants at the different measurements (mean ± standard deviation).

	Measurement 1 (*n* = 50)	Measurement 2 (*n* = 50)	Measurement 3 (*n* = 45)
Male/Female	15/35	15/35	13/32
Age in years	33.3 ± 8.6	33.3 ± 8.6	33.1 ± 8.5
Size in meters	1.69 ± 0.09	1.69 ± 0.09	1.7 ± 0.09
Weight in kilograms	68.8 ± 13.8	68.8 ± 13.8	69 ± 13.3

**Table 2 sensors-24-03699-t002:** Intertester reliability.

	M1 (cm) Mean ± SD	M2 (cm) Mean ± SD	ICC (95% CI)	SEM (cm) (SEM %)	SDC (cm) (SDC %)
Mean (3 jumps)	14.5 ± 4.1	14.5 ± 4.1	0.99 (0.98 to 0.99)	0.46 (3)	1.28 (9)
Max (3 jumps)	15.3 ± 4.4	15.3 ± 4.3	0.97 (0.94 to 0.98)	0.80 (5)	2.21 (14)

Note: M1: Measurement 1; M2: Measurement 2; ICC: intraclass coefficient; SEM: standard error of measurement, SDC: smallest detectable change; CI: confidence interval; SD: standard deviation.

**Table 3 sensors-24-03699-t003:** Intratester reliability.

	M1 (cm) Mean ± SD	M3 (cm) Mean ± SD	ICC (95% CI)	SEM (cm) (SEM %)	SDC (cm) (SDC %)
Mean (3 jumps)	14.2 ± 4.0	14.6 ± 4.6	0.96 (0.93 to 0.98)	1.47 (10)	4.08 (28)
Max (3 jumps)	15.0 ± 4.3	15.7 ± 5.1	0.82 (0.70 to 0.90)	2.60 (16)	7.21 (45)

Note: M1: Measurement 1; M3: Measurement 3; ICC: intraclass coefficient; SEM: standard error of measurement, SDC: smallest detectable change; CI: confidence interval; SD: standard deviation.

**Table 4 sensors-24-03699-t004:** Vailidity of agreement of the mean value of the CMJ measured by means of the Orthelligent Pro sensor and the force plate (using ground reaction force).

	Orthelligent (cm) Mean ± SD	Force Plate (cm) Mean ± SD	Pearson (r) (95% CI)	Mean Difference (cm) (95% CI)	*p* Value (*t*-Test)
Mean (3 jumps)	14.5 ± 4.1	10.7 ± 4.1	0.96 (0.94 to 0.98)	3.84 (3.54 to 4.15)	<0.001
Max (3 jumps)	15.3 ± 4.4	11.4 ± 4.3	0.94 (0.90 to 0.97)	3.91 (3.48 to 4.33)	<0.001

Note: CI = confidence interval; SD = standard deviation; r = Pearson correlation coefficient.

## Data Availability

The data presented in this study are available on request from the corresponding author.

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
