# Peer review of "Validity and Reliability of the Orthelligent Pro Sensor for Measuring Single-Leg Vertical Jump Height in Healthy Athletic Adults"

_sensors, 2024, doi:10.3390/s24123699_

Round 1
Reviewer 1 Report
Comments and Suggestions for Authors
The work is devoted to determining the reliability of the Orthelligent Pro device for measuring vertical jump height in humans (athletes in particular). To achieve this goal, the authors recruited 50 people aged 18 to 65 years.
At the same time, it is surprising that the assessment of the device for validity and reliability was carried out on changes in the kinematic characteristics of only vertical jumps such as a counter-jump. Another surprise is that the authors used a one-legged jump as a test, since such a jump is an unnatural movement. Countermovement jumping is a movement that involves a stretch shortening cycle that allows the body to store and redirect energy through and an eccentric movement quickly followed by a concentric movement. Because of the stretch shortening cycle, more force and power can be performed during the concentric phase of the jump than if no eccentric was performed. This can easily be seen when comparisons are made between squat jumps and сountermovement jumping. Jumping, as is known, in general is an extremely complex coordination movement, and a jump on one leg no longer requires great coordination of both intermuscular and nervous regulation. Therefore, when assessing the validity, reliability, and reproducibility of data, one cannot limit oneself only to changes in the kinematic parameters of jumps; physiological studies are required, which are absent in the work.
Reviewer 2 Report
Comments and Suggestions for Authors
Dear Authors,
First of all, I would like to congratulate you on having carried out this research, which, although of limited scientific and clinical impact, is interesting and presents novel objectives.
Unfortunately, the manuscript has shortcomings in its overall writing that need to be addressed before its possible publication in this Journal.
ABSTRACT: Abbreviations are discouraged in this section. They should be removed. Results should be transmitted after a = (not after :).
Keywords are very unfortunately chosen (they do not belong to MeSH and repeat terms that appear in the title).
INTRODUCTION: There is an abuse in the use of abbreviations which are then not used throughout the text even five times.
The adequacy and justification of this research is poor and lacks the formal rigour required of a scientific text. This section should be rewritten, expanded and referenced more intensively (with more and more recent bibliographical references).
METHODS: This section also makes very limited use of bibliographical references.
The calculation of the sample size needed to achieve the necessary statistical confidence is missing. As well as the subsequent calculation of the effect size of the final sample analysed (and other statistical techniques applied for data analysis).
RESULTS: It is sufficient to transmit descriptive results with a single decimal figure. The zero as the last decimal figure does not mean anything (it must be eliminated).
DISCUSSION: The weakest section. Instead of arguing and legitimizing the results obtained, invest the space in REPEATING the results already presented. There is no argument about the coherence, meaning or clinical impact of the results obtained.
CONCLUSIONS: Lacking an adequate Discussion, the generation of conclusions is also weak and lacks adequately justified clinical impact.
Kind regards
Reviewer 3 Report
Comments and Suggestions for Authors
Dear authors,
below are comments and suggestions to improve your study so that it can be considered eligible for publication.
Introduction
Lines 14-17: it is suggested to combine the 3 phrases in one.
Line 16: it is suggested to specify if the authors are referring to validity and reliability of sensor + data analysis software = system, or for example only sensor.
Line 41: 3D motion analysis refers to kinematics not kinetics. A couple of references are needed here about jump height estimation via kinematic data.
Line 42: I disagree with the wording of this phrase, because F/P serve to record GRF along with time, and from the force-time plot, flight time is determined based on which jump height will be estimated, provided that some assumptions are made (for instance, the subject took off and landed at the exact same body position, otherwise the projectile equations will not correctly give that t_ascent = t_descent).
Line 58-59: The force plate should be defined as the criterion method and is suggested to be used throughout the manuscript.
Lines 60-63: It does not become clear from the introduction how the authors formed their hypotheses. Reference 10 redirects the reader to the sensor's website where several previous articles that have used this system can be found. So, it is first recommended that a couple of references are added, may be at the end of phrase "The tests available on the accompanying ‟Orthelligent Pro App‟ comply with current return to sport standards".
Then, the authors could add a paragraph showing the benefits of sensors for assessment of jump height from previously published work.
Materials and Methods
Line 65: monocentric?
Lines 86-87: How was the thickest part assessed? Did the authors assessed the participant's dominant leg? If leg dominance was not assessed, could the authors jusify why not? was the sensor placed on the same side for each participant?
Line 87: what is the sampling frequency of acceleration data of the sensor? If it cannot record at 500Hz like the F/P's sampling frequency, the authors need to describe the action taken by them to address this issue. No such relevant information is shown in section 2.6.
Line 125: "jumped as high as possible into a vertical position" - unless a videocamera recorded their jump trials, the authors have no way to verify that. It is recommended that they delete the "into a vertical position".
Figure 2: The Figure shows the participant on the F/P while wearing the sensor. So, it is suggested to describe the Figure appropriately.
Section 2.6 Data analysis: the authors should provide the equations that Orth.Pro and F/P derived-data use to estimate jump height. There can be considerable misunderstanding by just stating like in Table 4 "...value of CMJ .... by means of the force plate (using GRF)". So, was jump height calculated from the fight phase or the propulsion phase of the GRF data?
Lines 142-143: "Reliability was assessed based on the two calculation methods" = it is suggested that another wording is used, because the study focuses on reliability and validity of the system based on the two different methods the F/P-derived data estimate jump height (considered as the criterion method) and the respective method the Orth.Pro does. On the contrary, the above is not a calculation method rather it refers to how data were treated off-line. Also, I am confused as well; do the Bland-Altman plots refer to the differences between examiners based on jump height as the mean value of 3 trials and as the value of the best trial or do they refer to difference of jump height as estimated by Orth.Pro and the F/P method?
Lines 147-148: this line is unnecessary and incorrect. Please remove it and the reference nr.12 should be placed at the end of the paragraph's 2nd phrase.
Lines 151-152: the reference from where the SDC statistic was found needs to be added.
Lines 153-156: the authors should provide cut off values for different level of the Pearson's correlation analysis.
Results
The Bland-Altman plots are not visually appropriate. Definition of both axes is not visible. Analysis of all these figures is highly recommended to be improved.
Line 182: The Figure title and legend is located below the figure. Do the same throughout the manuscript.
Discussion
Lines 262-264: How did the authors came to such a statement? They did not compare between the two methods.
Lines 308-310: here, calculations about take-off velocity are reported for the first time. I cannot see the relevence but then, may be the authors first calculated take-off velocity and then use the values to estimate jump height. However, this is a speculation since important information are missing with regard to data analysis (please see earlier comment).
Conclusions
Lines 342-345: please remove these lines. They cannot serve as a conclusion, they refer to the results.
Section reference = at the end of some references there is this "From NLM". Please remove it.
Line 392-393: It is suggested that the authors consult the following since they had 3 trials per participant.
Bland, J. M., & Altman, D. G. (2007). Agreement between methods of measurement with multiple observations per individual. Journal of biopharmaceutical statistics, 17(4), 571–582. https://doi.org/10.1080/10543400701329422
Also delete the word "scopus".
General comment: The title, abstract and discussion should be re-viewed once all the previous issues have been addressed.
Comments on the Quality of English LanguageNo comments.
Round 2
Reviewer 1 Report
Comments and Suggestions for Authors
The authors have made some corrections. However, the main question remains.
The authors point out: “Given this error, physical therapists may use Ortel-26. ligent Pro as an effective and reliable device for measuring jump heias well as in other vertical jumps and for vertical jumps such as squat jumps and fall jumps?.
There is no answer and the authors do not raise this question. This is the main question when it comes to using a new device to assess the biomechanical efficiency of muscles during vertical jump.
Reviewer 2 Report
Comments and Suggestions for Authors
Dear Authors,
Once again, congratulations on this very interesting research. After implementing the suggested corrections, I now consider that the manuscript should be accepted for publication in this Journal.
Kind regards
Reviewer 3 Report
Comments and Suggestions for Authors
Dear authors,
Your revised manuscript has definitely improved the quality of your work. However, there are still issues to be considered before the manuscript can be ready for publication. Please see below.
General comment
Since the authors use the brand name of the sensor, it would make more sense if they do the same for the F/P. So, it is suggested that they write "Kistler F/P" throughout the manuscript. Or alternatively "Quattro Jump F/P". Also, no need to write after F/P that it relates to ground reaction forces, since that is what a F/P measures, there can be no confusion about it.
Revise the whole manuscript to make sure there is a space between the end of a sentence and the reference number in brackets.
Abstract
Lines 22-23: It is suggested that the reference to the Bland-Altman plots shown in the Results be about the fact that they are also used to analyse the agreement between the 2 methods in the calculation of jump height. This might be more appropriate for the study's purpose since one of the purposes is to examine the validity of the sensor. Therefore, it is recommended that the authors adjust that phrase appropriately.
Introduction
All the paragraphs in this section apart from the last two, are not presented in a way that would justify the purpose and even more the study's hypothesis. In its current form, the introduction reads like pieces of information here and there with no structured connection between them. The purpose is to examine the reliability and validity of this need sensor by means of a test that provides useful information for athletes. So, in my opinion the introduction should first focus on data about the reliability and validity of sensors previously used in sports with reference to jump height, and from that point on the introduction could be about a) significance of jump height for sports, b) relevance with athlete's return to sport level, c) F/P as the golden method to derive parameters from the GRF data and calculate jump height and last about their choice to use a single-legged CMJ test to evaluate jump height.
The hypotheses that the authors present is not substantiated in the introduction's current form.
Line 78: delete the word "monocentric".
Lines 87-88: based on the information here the participants are physically active persons. However, it is suggested that the physical activity level be defined better, since active at least once a week can have significant differences among those 50 persons. Try to provide information about intensity level, duration of session, frequency so that readers have a clear idea.
Lines 90-91: "known or suspected non-compliance with the protocol" = this criterion would make sense if this was a longitudinal study with various measures. It does make sense to put it otherwise. It is suggested that is removed.
Line 90: the informed consent is not an inclusion criterion. it needs to be removed.
Line 100: the calf of their jumping leg is vague. PLease define if the dominant leg was used or not, and if this was not the case justify why you chose not to use the DOM leg for the single-leg CMJ.
Line 103: Personally, I would rather see a figure illustrating how measured and analyzed data appear on the app's environment compared to Figure 2, which can be nicely described inside the manuscript.
Line 109: The calculation of jump height from F/P needs a bit more work. The authors are encouraged to refer to Eagles, Sayers, et al al. 2015 study they use since using the propulsion phase means to integrate the Fz data and for that purpose, the start and the end point of the time period of the integration of Fz data is needed to be specified.
So, the phrase "via the propulsion phase by integrating the ground reaction force" needs to be corrected.
Lines 130-135, Figure 1. The box to depict the validity part of the study does not show that 3 jumps were simultaneously recorded with the sensor and the kistler F/P. Please address this issue.
Line 173: it is not sociodemographic rather than anthropometric data.
Line 195: Legend is typically what follows the Figure. At the bottom of the Table, typically one could write Note.
Line 214: there is no Legend below a Table, refer to earlier similar comment.
Line 235: same comment as in line 214.
Lines 238-243: Figure 5: the same comment as my general comment about how to refer to using the F/P as the golden standard method.
Discussion: overall comment.
Significant part of the discussion is spent in repeating the results. If the authors cannot escape from doing so, they could do it in the 1st paragraph of the discussion where they could write if their hypotheses were verified or rejected. It is highly recommended that attempt to explain their results by a) the fact that the sensor uses an accelerometer, from which probably it derives the flight phase vs. the direct recording of the GRF data by means of the F/P, b) the method of jump height calculation, since if one uses the flight phase there are certain assumptions to made (equations of constant acceleration) as compared with integrating the FZ in the propulsion phase from the recorder GRF data, c) the task itself: the single-leg CMJ is a challenging task and combined with the wide age range of participants, it does not come as a surprise that the mean of 3 jumps gave more reliable data compared to the maximum of the 3 jumps.
Further, once the authors make clear if participants jumped with their dominant leg or not, this could be help them interpret the differences between sensor and F/P with regards to how neuromuscular coordination (with dominant or non-dominant leg) changes performance and how this can have an impact on the method of jump height calculation (flight phase by sensor vs. integration of propulsion phase by F/P).
Lines 300-302: The references nr.31 and 32 used by the authors here have no relation with a) the outcome measure (jump height), or b) the participants (physically active persons here vs participants with either shoulder pain (ref.nr.31) or myopathy (ref.nr.32). The authors are advised to remove these references and replace with appropriate ones.
Conclusions: this section could be shortened.
Comments on the Quality of English LanguageNo comments here.
Round 3
Reviewer 3 Report
Comments and Suggestions for Authors
Dear authors,
Congratulations for your effort in addressing my comments. The revised manuscript is significantly improved. There are some minor issues to be addressed still, please see below.
Table 1: there is no point of presenting anthropometric data by gender, since this effect was not examined. It is suggested to present data based on the number of subjects in each measurement. Measure 1 = 50, measure 2 = 46, measure 3 = 45.
Correct legend of Table 1: not demographic but anthropometric.
Lines 187, 205, 223: delete the word "Legend". The description constitutes the legend of a figure. we don't write it.
Line 247: "mean difference of 0.3 cm -0.7 cm" = is there something missing here? may be it is 0.3 +/-0.7 cm?
Lines 276-277: unless I missed something, there are no data comparing between sides. So, I am confused about the reference to Limb Symmetry Index measurements. May be the authors could explain what do they mean.
Line 273: correct "free" to "three".
Line 281: correct to "...by being active...".
Comments on the Quality of English LanguageSee comments below.
